

# Constraining new physics from Higgs measurements with Lilith: update to LHC Run 2 results

**Sabine Kraml**[1*]**, Tran Quang Loc**[2,3]**, Dao Thi Nhung**[2] **and Le Duc Ninh**[2]

**1** Laboratoire de Physique Subatomique et de Cosmologie, Université Grenoble-Alpes, CNRS/IN2P3, 53 Avenue des Martyrs, F-38026 Grenoble, France
**2** Institute For Interdisciplinary Research in Science and Education, ICISE, 590000 Quy Nhon, Vietnam
**3** VNUHCM-University of Science, 227 Nguyen Van Cu, Dist. 5, Ho Chi Minh City, Vietnam

⋆ sabine.kraml@lpsc.in2p3.fr

## Abstract

Lilith is a public Python library for constraining new physics from Higgs signal strength measurements. We here present version 2.0 of Lilith together with an updated XML database which includes the current ATLAS and CMS Run 2 Higgs results for 36 fb$^{-1}$. Both the code and the database were extended from the ordinary Gaussian approximation employed in Lilith-1.1 to using variable Gaussian and Poisson likelihoods. Moreover, Lilith can now make use of correlation matrices of arbitrary dimension. We provide detailed validations of the implemented experimental results as well as a status of global fits for reduced Higgs couplings, Two-Higgs-doublet models of Type I and Type II, and invisible Higgs decays. Lilith-2.0 is available on GitHub and ready to be used to constrain a wide class of new physics scenarios.


## Contents



## 1  Introduction

The LHC runs in 2010–2012 and 2015–2018 have led to a wealth of experimental results on the 125 GeV Higgs boson. From this emerges an increasingly precise picture of the various Higgs production and decay processes, and consequently of the Higgs couplings to the other particles of the Standard Model (SM), notably gauge bosons and third generation fermions. With all measurements so far agreeing with SM predictions, this poses severe constraints on scenarios of new physics, in which the properties of the observed Higgs boson could be affected in a variety of ways.

Assessing the compatibility of a non-SM Higgs sector with the ATLAS and CMS results requires to construct a likelihood, which is a non-trivial task. While this is best done by the experimental collaborations themselves, having at least an approximate global likelihood is very useful, as it allows theorists to pursue in-depth studies of the implications for their models. For this reason, the public code Lilith [1] (see also [2]) was created, making use of the Higgs signal strength measurements published by ATLAS and CMS, and the Tevatron experiments.[1] In this paper, we present version 2.0 of Lilith together with an updated database which includes the current published ATLAS and CMS Run 2 Higgs results for 36 fb$^{-1}$.

Compared to HiggsSignals [4], which is written in Fortran90 and uses the signal strengths for individual experimental categories with their associated efficiencies as well as Simplified Template Cross Section (STXS) [5,6] results, Lilith is a light-weight Python library that uses as a primary input signal strength results

$$\mu(X, Y) \equiv \frac{\sigma(X)\,\mathrm{BR}(H \to Y)}{\sigma^{\mathrm{SM}}(X)\,\mathrm{BR}^{\mathrm{SM}}(H \to Y)}, \tag{1}$$

in which the fundamental production and decay modes are unfolded from experimental categories. Here, the main production mechanisms $X$ are: gluon fusion (ggH), vector-boson fusion (VBF), associated production with an electroweak gauge boson (WH and ZH, collectively denoted as VH) and associated production with top quarks, mainly ttH but also tH. The main decay modes $Y$ accessible at the LHC are $H \to \gamma\gamma$, $H \to ZZ^* \to 4\ell$, $H \to WW^* \to 2\ell2\nu$, $H \to b\bar{b}$ and $H \to \tau\tau$ (with $\ell \equiv e, \mu$).[2]

The signal strength framework is based on the narrow-width approximation and on the assumption that new physics results only in the scaling of SM Higgs processes.[3] This makes it possible to combine the information from various measurements and assess the compatibility

---

[1]For a discussion of the use and usability of signal strength results, and recommendations on their presentation, see [3].

[2]When $\mu(X, Y)$ is not directly available, $\mu = \sum \mathrm{eff}_{X,Y} \times \mu(X, Y)$ is used, introducing appropriate efficiency factors $\mathrm{eff}_{X,Y}$. For inclusive combinations of production channels for the same $Y$, the efficiencies become $\mathrm{eff}_X = \sigma^{\mathrm{SM}}(X) / \sum \sigma^{\mathrm{SM}}(X)$.

[3]In other words, the Lagrangian has the same tensor structure as the SM, see e.g. the discussion in [3].

of given scalings of SM production and/or decay processes from a global fit to the Higgs data. This framework is very powerful as it can be used to constrain a wide variety of new physics models, see for example [7] and references therein.

For a proper inclusion of the recent Run 2 results from ATLAS and CMS, several improvements were necessary in Lilith. Concretely, in order to treat asymmetric uncertainties in a better way, we have extended the parametrisation of the likelihood to Gaussian functions of variable width ("variable Gaussian") as well as generalised Poisson functions. Moreover, Lilith can now make use of correlation matrices of arbitrary dimension. We have also added the tH and the gluon-initiated ZH production modes, and corrected some minor bugs in the code.

Results given in terms of signal strengths can be matched to new physics scenarios with the introduction of factors $C_X$ and $C_Y$ that scale the amplitudes for the production and decay of the SM Higgs boson, respectively, as

$$\mu(X, Y) = \frac{C_X^2 C_Y^2}{\sum_Y C_Y^2 \, \mathrm{BR}^{\mathrm{SM}}(H \to Y)} \qquad (2)$$

for the different production modes $X \in (\mathrm{ggH}, \mathrm{VBF}, \mathrm{WH}, \mathrm{ZH}, \mathrm{ttH}, \dots)$ and decay modes $Y \in (\gamma\gamma, ZZ^*, WW^*, b\bar{b}, \tau\tau, \dots)$, where the sum runs over all decays that exist for the SM Higgs boson. The factors $C_X$ and $C_Y$ can be identified to (or derived from) reduced couplings appearing in an effective Lagrangian. Following [8] and subsequent publications, we employ the notation

$$\mathcal{L} = g \left[ C_W m_W W^\mu W_\mu + C_Z \frac{m_Z}{\cos\theta_W} Z^\mu Z_\mu - \sum_f C_f \frac{m_f}{2 m_W} f \bar{f} \right] H, \qquad (3)$$

where $C_{W,Z}$ and $C_f$ ($f = t, b, c, \tau, \mu$) are bosonic and fermionic reduced couplings, respectively. In the limit where all reduced couplings go to 1, the SM case is recovered. In addition to these tree-level couplings, we define the loop-induced couplings $C_g$ and $C_\gamma$ of the Higgs to gluons and photons, respectively. If no new particles appear in the loops, $C_g$ and $C_\gamma$ are computed from the couplings in Eq. (3) following the procedure established in [9]. Alternatively, $C_g$ and $C_\gamma$ can be taken as free parameters. Apart from the different notation, this is equivalent to the so-called "kappa framework" of [9]. Finally note that often a subset of the $C$'s in Eq. (3) is taken as universal, in particular $C_V \equiv C_W = C_Z$ (custodial symmetry), $C_U \equiv C_t = C_c$ and $C_D \equiv C_b = C_\tau = C_\mu$ like in the Two-Higgs-doublet model (2HDM) of Type II, or $C_F \equiv C_U = C_D$ as in the 2HDM of Type I.

Last but not least, while the signal strength framework in principle requires that the Higgs signal be a sum of processes that exist for the SM Higgs boson, decays into invisible or undetected new particles, affecting only the Higgs total width, can be accounted for through

$$\mu(X, Y) \to [1 - \mathrm{BR}(H \to \mathrm{inv.}) - \mathrm{BR}(H \to \mathrm{undet.})] \, \mu(X, Y), \qquad (4)$$

without spoiling the approximation.

The rest of the paper is organised as follows. We begin the discussion of novelties in Lilith-2.0 by presenting the extended XML format for experimental input in Section 2. This is followed by details on the calculation of the likelihood in Section 3. The inclusion of new production channels is described in Section 4. The Run 2 results included in the new database and their validation are presented in Section 5. In Section 6 we then give an overview of the current status of Higgs coupling fits with Lilith-2.0. We conclude in Section 7. Appendix A contains additional material illustrating the importance of various improvements discussed throughout the paper.

It is important to note that this paper is not a standalone documentation of `Lilith-2.0`. Instead, we present only what is new with respect to `Lilith-1.1`. For everything else, including instructions how to use the code, we refer the reader to the original manual [1].

## 2 Extended XML format for experimental input

In the `Lilith` database, every single experimental result is stored in a separate XML file. This allows to easily select the results to use in a fit, and it also makes maintaining and updating the database rather easy.

The root tag of each XML file is `<expmu>`, which has two mandatory attributes, `dim` and `type` to specify the type of signal strength result. Production and decay modes are specified via `prod` and `decay` attributes either directly in the `<expmu>` tag or in the efficiency `<eff>` tags. The latter option allows for the inclusion of different production and decay modes in one XML file. Additional (optional) information can be provided in `<experiment>`, `<source>`, `<sqrts>`, `<CL>` and `<mass>` tags. Taking the $H \to \gamma\gamma$ result from the combined ATLAS and CMS Run 1 analysis [10] as a concrete example, the structure is

```
<expmu decay="gammagamma" dim="2" type="n">
  <experiment>ATLAS-CMS</experiment>
  <source type="publication">CMS-HIG-15-002; ATLAS-HIGG-2015-07</source>
  <sqrts>7+8</sqrts>
  <mass>125.09</mass>
  <CL>68%</CL>

  <eff axis="x" prod="ggH">1.</eff>
  <eff axis="y" prod="VVH">1.</eff>
  <!-- (...) -->
</expmu>
```

where `<!-- (...) -->` is a placeholder for the actual likelihood information. For a detailed description, we refer to the original `Lilith` manual [1]. In the following, we assume that the reader is familiar with the basic syntax.

So far, the likelihood information could be specified in one or two dimensions in the form of [1]: 1D intervals given as best fit with $1\sigma$ error; 2D likelihood contours described as best fit point and parameters a, b, c which parametrise the inverse of the covariance matrix; or full likelihood information as 1D or 2D grids of $-2\log L$. The first two options, 1D intervals and 2D likelihood contours, declared as `type="n"` in the `<expmu>` tag, employ an ordinary Gaussian approximation; in the 1D case, asymmetric errors are accounted for by putting together two one-sided Gaussians with the same mean but different variances, while the 2D case assumes symmetric errors. This does does not always allow to describe the experimental data (i.e. the true likelihood) very well.

In order to treat asymmetric uncertainties in a better way, we have extended the XML format and likelihood calculation in `Lilith` to Gaussian functions of variable width ("variable Gaussian") as well as generalized Poisson functions [11]. The declaration is `type="vn"` for the variable Gaussian or `type="p"` for the Poisson form in the `<expmu>` tag. Both work for 1D and 2D data with the same syntax. Moreover, in order to make use of the multi-dimensional correlation matrices which both ATLAS and CMS have started to provide, we have added a new XML format for correlated signal strengths in more than two dimensions. This can be used with the ordinary or variable Gaussian approximations for the likelihood. In the following we give explicit examples for the different possibilities.

## 1D likelihood parametrisation

For 1D data, the format remains the same as in [1]. For example, a signal strength $\mu(ZH, b\bar{b}) = 1.12^{+0.50}_{-0.45}$ is implemented as

```
<eff prod="ZH">1.</eff>
<bestfit>1.12</bestfit>
<param>
  <uncertainty side="left">-0.45</uncertainty>
  <uncertainty side="right">0.50</uncertainty>
</param>
```

The `<bestfit>` tag contains the best-fit value, while the `<uncertainty>` tag contains the left (negative) and right (positive) $1\sigma$ errors.[4] The choice of likelihood function is done by setting `type="n"` for an ordinary, 2-sided Gaussian (as in `Lilith-1.1`); `type="vn"` for a variable Gaussian; or `type="p"` for a Poisson form in the `<expmu>` tag.

## 2D likelihood parametrisation

For `type="vn"` and `type="p"`, signal strengths in 2D with a correlation are now described in an analogous way as 1D data. For example, $\mu(ggH, WW) = 1.10^{+0.21}_{-0.20}$ and $\mu(VBF, WW) = 0.62^{+0.36}_{-0.35}$ with a correlation of $\rho = -0.08$ can be implemented as

```
<expmu decay="WW" dim="2" type="vn">

  <eff axis="x" prod="ggH">1.0</eff>
  <eff axis="y" prod="VBF">1.0</eff>

  <bestfit>
    <x>1.10</x>
    <y>0.62</y>
  </bestfit>

  <param>
    <uncertainty axis="x" side="left">-0.20</uncertainty>
    <uncertainty axis="x" side="right">+0.21</uncertainty>
    <uncertainty axis="y" side="left">-0.35</uncertainty>
    <uncertainty axis="y" side="right">+0.36</uncertainty>
    <correlation>-0.08</correlation>
  </param>
</expmu>
```

Here, the `<eff>` tag is used to declare the x and y axes, specified by their production and/or decay channels together with the corresponding efficiencies. The `<bestfit>` tag specifies the location of the best-fit point in the (x,y) plane. The `<uncertainty>` tags contain the left (negative) and right (positive) $1\sigma$ errors for the x and y axes, and finally the `<correlation>` tag specifies the correlation between x and y. The choice of likelihood function is again done by setting `type="vn"` or `type="p"` in the `<expmu>` tag.

To ensure backwards compatibility, `type="n"` however still requires the tags `<a>`, `<b>`, `<c>` to give the inverse of the covariance matrix instead of `<uncertainty>` and `<correlation>`, see [1].

---

[4]The values in the `<uncertainty>` tag can be given with or without a sign.

**Multi-dimensional data**

For correlated signal strengths in more than 2 dimensions, a new format is introduced. We here illustrate it by means of the CMS result [12], which has signal strengths for 24 production and decay mode combinations plus a 24 × 24 correlation matrix. First, we set `dim="24"` and label the various signal strengths as axes d1, d2, d3, … d24:[5]

```
<expmu dim="24" type="vn">
  <eff axis="d1" prod="ggH" decay="gammagamma">1.0</eff>
  <eff axis="d2" prod="ggH" decay="ZZ">1.0</eff>
  <eff axis="d3" prod="ggH" decay="WW">1.0</eff>
  ...
  <eff axis="d24" prod="ttH" decay="tautau">1.0</eff>
```

The best-fit values for each axis are specified as

```
  <bestfit>
    <d1>1.16</d1>
    <d2>1.22</d2>
    <d3>1.35</d3>
    ...
    <d24>0.23</d24>
  </bestfit>
```

The `<param>` tag then contains the uncertainties and correlations in the form

```
  <param>
    <uncertainty axis="d1" side="left">-0.18</uncertainty>
    <uncertainty axis="d1" side="right">+0.21</uncertainty>
    <uncertainty axis="d2" side="left">-0.21</uncertainty>
    <uncertainty axis="d2" side="right">+0.23</uncertainty>
    ...
    <uncertainty axis="d24" side="left">-0.88</uncertainty>
    <uncertainty axis="d24" side="right">+1.03</uncertainty>

    <correlation entry="d1d2">0.12</correlation>
    <correlation entry="d1d3">0.16</correlation>
    <correlation entry="d1d4">0.08</correlation>
    ...
    <correlation entry="d23d24">0</correlation>
  </param>
</expmu>
```

This will also work for `type="n"`, see Eq. (9) in the next section.

   We are aware that having different formats for 2 and more than 2 dimensions is not necessary in principle. Nonetheless we chose to treat the 2D case separately (with axis tags `"x"` and `"y"` instead of `"d1"` and `"d2"`) to stay as close as possible to what was done in `Lilith-1.1`. This may change in future versions.

---

[5]The `<experiment>`, `<source>`, `<sqrts>`, etc. tags are omitted for brevity.

# 3 Likelihood calculation

The statistical procedure used in `Lilith` was described in detail in [1]. The main quantity given as an output is the log-likelihood, $-2\log L$, evaluated in `computelikelihood.py` from the information given in the XML database. In this section, we explain how $-2\log L$ is computed for `type="vn"` (variable Gaussian) and `type="p"` (Poisson) introduced in the previous section. For the old implementation of the ordinary Gaussian (`type="n"`), we refer the reader to [1].

## 3.1 Variable Gaussian

As shown in [11], a Gaussian function of variable width can be a good choice to deal with asymmetric uncertainties. We use the version linear in the variance, described as "Variable Gaussian (2)" in Section 3.6 of [11]. In the 1D case, the likelihood is then given by

$$-2\log L(\mu) = \frac{(\mu - \hat{\mu})^2}{\sigma^+\sigma^- + (\sigma^+ - \sigma^-)(\mu - \hat{\mu})}, \tag{5}$$

where $\hat{\mu}$ denotes the best-fit signal strength, and $\sigma^-$ and $\sigma^+$ are absolute values of the left and right uncertainties at 68.3% CL, respectively. If not stated otherwise, these notations are used for the entire section. For $\sigma^+ = \sigma^-$, the symmetric case is obtained. The variable Gaussian form however has a singularity at $\mu = \hat{\mu} - (\sigma^+\sigma^-)/(\sigma^+ - \sigma^-)$, which can lead to numerical issues, although in practice this usually happens for $\mu$'s outside the range of interest (or reduced couplings outside their physically meaningful range). The numerical stability can also be problematic when $\sigma^- \to 0$, in which case it may be better to use the ordinary 2-sided Gaussian implementation, in particular for 1D data; this has to be decided case-by-case upon validation. In any case, `Lilith` issues an error message whenever a numerical problem occurs.

In the case of $n$-dimensional data ($n > 1$), we use the correlations given by the experimental collaboration, if available, together with the best fit points and the left and right uncertainties at 68.3% CL. When results are given in terms of 2D contour plots, we can also use the variable Gaussian form to numerically determine the best-fit point, uncertainties and their correlation, if not given explicitly by the experimental collaboration. For the $n$ dimensional signal strength vector $\boldsymbol{\mu} = (\mu_1, \ldots, \mu_n)$, the likelihood reads

$$-2\log L(\boldsymbol{\mu}) = (\boldsymbol{\mu} - \hat{\boldsymbol{\mu}})^T C^{-1} (\boldsymbol{\mu} - \hat{\boldsymbol{\mu}}), \tag{6}$$

where the best fit point $\hat{\boldsymbol{\mu}} = (\hat{\mu}_1, \ldots, \hat{\mu}_n)$ and the covariance matrix is constructed from the correlation matrix $\rho$ as

$$C = \Sigma(\boldsymbol{\mu}).\rho.\Sigma(\boldsymbol{\mu}), \quad \Sigma(\boldsymbol{\mu}) = \text{diag}(\Sigma_1, \ldots, \Sigma_n), \tag{7}$$

with

$$\Sigma_i = \sqrt{\sigma_i^+\sigma_i^- + (\sigma_i^+ - \sigma_i^-)(\mu_i - \hat{\mu}_i)}, \quad i = 1, \ldots, n. \tag{8}$$

Here the $\sigma_i^-$ and $\sigma_i^+$ are the left and right uncertainties at 68.3% CL of the $i$th combination of production and/or decay channels, respectively. For multi-dimensional data in the ordinary Gaussian approximation, the relation between covariance matrix and the correlation matrix becomes

$$C = \frac{1}{4}[\boldsymbol{\sigma}^+ + \boldsymbol{\sigma}^-].\rho.[\boldsymbol{\sigma}^+ + \boldsymbol{\sigma}^-], \tag{9}$$

where $\boldsymbol{\sigma}^+ = \text{diag}(\sigma_1^+, \ldots, \sigma_n^+)$ and $\boldsymbol{\sigma}^- = \text{diag}(\sigma_1^-, \ldots, \sigma_n^-)$.

## 3.2 Generalised Poisson

As an alternative to the variable Gaussian, a generalised Poisson form can be used for 1D and 2D results. For the 1D case, the likelihood is implemented according to Section 3.4, "Generalised Poisson", of [11] as

$$\log L(\mu) = -\nu\gamma(\mu - \hat{\mu}) + \nu\log\left[1 + \gamma(\mu - \hat{\mu})\right], \tag{10}$$

where $\gamma$ and $\nu$ are determined numerically from the equations

$$\frac{1 - \gamma\sigma^-}{1 + \gamma\sigma^+} = e^{-\gamma(\sigma^+ + \sigma^-)}, \quad \nu = \frac{1}{2(\gamma\sigma^+ - \log(1 + \gamma\sigma^+))}. \tag{11}$$

More concretely, $\gamma$ is determined from the expression on the left by bifurcation between $\gamma = 0$ and $\gamma = 1/\sigma^-$ and then inserted in the expression on the right to compute $\nu$.

For the 2D case, we use the conditioning bivariate Poisson distribution described in [13], that has no restriction on the sign and magnitude of the correlation $\rho$. Here the joint distribution is a product of a marginal and a conditional distribution. The decision of which channel belongs to the marginal or the conditional distribution is based on the validation plots. To illustrate the method, we assume that the data of the channel $X$ follows the marginal distribution, while data of the channel $Y$ belongs to the conditional distribution. The joint log-likelihood is then

$$\log L(\mu_X, \mu_Y) = \log L(\mu_X) + \log L(\mu_Y | \mu_X), \tag{12}$$

with

$$\log L(\mu_X) = -\nu_X\gamma_X(\mu_X - \hat{\mu}_X) + \nu_X\log\left[1 + \gamma_X(\mu_X - \hat{\mu}_X)\right], \tag{13}$$

and

$$\log L(\mu_Y | \mu_X) = f(\mu_X, \mu_Y) - f(\hat{\mu}_X, \hat{\mu}_Y) + \nu_Y\log\frac{f(\mu_X, \mu_Y)}{f(\hat{\mu}_X, \hat{\mu}_Y)}. \tag{14}$$

The function $f$ reads

$$f(a, b) = -\nu_Y\gamma_Y\left(b - \hat{\mu}_Y + \frac{1}{\gamma_Y}\right)\exp\left[\nu_X\alpha - (e^\alpha - 1)\nu_X\gamma_X(a - \hat{\mu}_X + \frac{1}{\gamma_X})\right], \tag{15}$$

where $\alpha$ is solved numerically from the correlation expression

$$\rho = \frac{\nu_X\nu_Y(e^\alpha - 1)}{\sqrt{\nu_X\nu_Y\left[1 + \nu_Y\left(e^{\nu_X(e^\alpha - 1)^2} - 1\right)\right]}}, \tag{16}$$

and the $\gamma_X, \nu_X$ and $\gamma_Y, \nu_Y$ are solutions of Eq. (11) for the $X$ and $Y$ channels, respectively.

# 4 New production channels

As mentioned in the introduction, we have also included a few new production channels. These are ZH production via gluon-gluon fusion (ggZH), Higgs production in association with a single top quark (tH), and Higgs production in association with a pair of bottom quarks (bbH).

For the ZH production mode, the original implementation in `Lilith` included only the $q\bar{q} \to ZH$ channel (qqZH). However, the loop-induced gluon-gluon fusion is not so small, about 14% of the total $pp \to ZH$ cross section at $\sqrt{s} = 13$ TeV, and should hence be taken into account. Indeed, both ATLAS and CMS have been always including the ggZH contribution in their fits. From version 2 onwards, the ZH signal in `Lilith` is also the combination of the $q\bar{q}$

and $gg$ initiated processes, with relative weights $\sigma_X^{\mathrm{SM}}/(\sum_X \sigma_X^{\mathrm{SM}})$. In terms of the scale factors of Eq. (2), this gives

$$C_{\mathrm{ZH}}^2 = \left(\sigma_{\mathrm{qqZH}}^{\mathrm{SM}} C_{\mathrm{qqZH}}^2 + \sigma_{\mathrm{ggZH}}^{\mathrm{SM}} C_{\mathrm{ggZH}}^2\right) / \left(\sigma_{\mathrm{qqZH}}^{\mathrm{SM}} + \sigma_{\mathrm{ggZH}}^{\mathrm{SM}}\right). \tag{17}$$

For the SM cross sections, the values given in [6, 14, 15] are used, where the qqZH cross section is calculated at the Next-to-Next-to-Leading Order (NNLO) QCD + Next-to-Leading Order (NLO) electroweak level, and the ggZH cross section is calculated at the NLO QCD level. The definitions of VH (ZH and WH) and VVH (ZH, WH and VBF) follow straightforwardly. The WH cross section is calculated at NNLO QCD and NLO electroweak accuracies, while the VBF cross section is calculated at approximate NNLO QCD and NLO electroweak accuracies. Further details are provided in [6, 14, 15].

The tH production also includes two contributions: $t$-channel $tHq$ production and $tHW$ production. The $s$-channel $tHq$ cross section is much smaller and hence not included. Interference effects between these channels are also neglected. At $\sqrt{s} = 13$ TeV, $tHq$ is dominant, contributing about 80% of the tH cross section. As above, tHq and tHW are combined to tH production with efficiencies calculated from the SM cross sections. Moreover, following the usage in the ATLAS analysis [16], a combination of tH and ttH named 'top' production is defined in an analogous way.[6] Thus

$$C_{\mathrm{tH}}^2 = \left(\sigma_{\mathrm{tHq}}^{\mathrm{SM}} C_{\mathrm{tHq}}^2 + \sigma_{\mathrm{tHW}}^{\mathrm{SM}} C_{\mathrm{tHW}}^2\right) / \left(\sigma_{\mathrm{tHq}}^{\mathrm{SM}} + \sigma_{\mathrm{tHW}}^{\mathrm{SM}}\right), \tag{18}$$

$$C_{\mathrm{top}}^2 = \left(\sigma_{\mathrm{tHq}}^{\mathrm{SM}} C_{\mathrm{tHq}}^2 + \sigma_{\mathrm{tHW}}^{\mathrm{SM}} C_{\mathrm{tHW}}^2 + \sigma_{\mathrm{ttH}}^{\mathrm{SM}} C_{\mathrm{ttH}}^2\right) / \left(\sigma_{\mathrm{tHq}}^{\mathrm{SM}} + \sigma_{\mathrm{tHW}}^{\mathrm{SM}} + \sigma_{\mathrm{ttH}}^{\mathrm{SM}}\right). \tag{19}$$

The value for $\sigma_{\mathrm{ttH}}^{\mathrm{SM}}$ is calculated at NLO QCD and NLO electroweak accuracies, while $\sigma_{\mathrm{tHq}}^{\mathrm{SM}}$ is calculated at NLO QCD level as given in [6,14,15]. $\sigma_{\mathrm{tHW}}^{\mathrm{SM}}$ has been calculated at leading order using MadGraph [17] with factorization and renormalization scales $\mu_F = \mu_R = (m_t + m_W + m_H)/2$ and the NNPDF30_lo_as_0130 PDF set [18]. Other input parameters are the same as in [6] section I.6 for $tH$ production. It is noted that the definition of an NLO cross section for the $tHW$ channel is not straightforward because of the interferences with the ttH channel, see [19] for a discussion on this issue.

For the sake of completeness and to prepare for future data, the bbH production mode has also been added. The implementation is straightforward, analogous to the ttH case.

Finally, since `Lilith` accepts reduced couplings as user input, the relations between the scaling factors and the reduced couplings are needed. These relations read, following the notation of [1],

$$C_{\mathrm{qqZH}}^2 = C_Z^2, \quad C_{\mathrm{ttH}}^2 = C_t^2, \quad C_{\mathrm{bbH}}^2 = C_b^2, \tag{20}$$

$$C_{\mathrm{ggZH}}^2 = a_t C_t^2 + a_b C_b^2 + a_Z C_Z^2 + a_{tb} C_t C_b + a_{tZ} C_t C_Z + a_{bZ} C_b C_Z, \tag{21}$$

$$C_{\mathrm{tHq}}^2 = e_t C_t^2 + e_W C_W^2 + e_{tW} C_t C_W, \tag{22}$$

$$C_{\mathrm{tHW}}^2 = f_t C_t^2 + f_W C_W^2 + f_{tW} C_t C_W, \tag{23}$$

where the coefficients $a_i$, $e_i$, and $f_i$, being the SM predictions, are provided in Tables 1 and 2. Note that for the time being fixed values for $m_H = 125$ GeV are used. Moreover, for the case $\sqrt{s} = 7$ TeV, the values at 8 TeV are used; the differences are negligible in the current approximations.

Finally, we note that the accuracies of the remaining SM predictions, including ggH cross sections and the Higgs branching fractions, are unchanged compared to the previous version of `Lilith`; in particular (N)NLO QCD corrections are included as explained in [1].

---

[6]Accordingly, `prod="tHq"`, `"tHW"`, `"tH"` and `"top"` can be used in the XML files in the database.

Table 1: $a_i$ coefficients for the ggZH signal strength, calculated at leading order QCD, taken from [20].

| $\sqrt{s}$ [TeV] | $a_t$ | $a_b$ | $a_Z$ | $a_{tb}$ | $a_{tZ}$ | $a_{bZ}$ |
|---|---|---|---|---|---|---|
| 8 | 0.372 | 0.0004 | 2.302 | 0.003 | −1.663 | −0.013 |
| 13 | 0.456 | 0.0004 | 2.455 | 0.003 | −1.902 | −0.011 |

Table 2: $e_i$ and $f_i$ coefficients for the tHq and tHW signal strengths, calculated at NLO QCD, taken from [20].

| $\sqrt{s}$ [TeV] | $e_t$ | $e_W$ | $e_{tW}$ | $f_t$ | $f_W$ | $f_{tW}$ |
|---|---|---|---|---|---|---|
| 8 | 2.984 | 3.886 | −5.870 | 2.426 | 1.818 | −3.244 |
| 13 | 2.633 | 3.578 | −5.211 | 2.909 | 2.310 | −4.220 |

## 5  ATLAS and CMS results included in the database update

In this section, we discuss the ATLAS and CMS Run 2 results included in the `Lilith` database. Most of this is based on the database release DB 19.06 (June 2019). Three ATLAS results (HIGG-2017-14, HIGG-2016-10 and HIGG-2018-54) were added to this during the peer-review process, so that the current latest version is DB 19.09 (September 2019). There is no other difference between DB 19.06 and DB 19.09. The validation plots in this section carry their original DB version number, that is 19.06 or 19.09, while the fit results in Section 6 are all for the complete DB 19.09.

### 5.1  ATLAS Run 2 results for 36 fb$^{-1}$

The ATLAS Run 2 results included in this release are summarised in Table 3 and explained in more detail below.

Table 3: Overview of ATLAS Run 2 results included in this release.

| mode | $\gamma\gamma$ | $ZZ^*$ | $WW^*$ | $\tau\tau$ | $b\bar{b}$ | $\mu\mu$ | inv. |
|---|---|---|---|---|---|---|---|
| ggH | [16] | [21] | [22] | [23] | – | [24] | – |
| VBF | [16] | [21] | [22] | [23] | [25] | [24] | [26] |
| WH | [16] | [21] | [27] | – | [28] | [24] | – |
| ZH | [16] | [21] | [27] | – | [28] | [24] | [29] |
| ttH | [16] | [21, 30] | [30] | [30] | [30, 31] | – | – |

$H \to \gamma\gamma$ **(HIGG-2016-21):** The ATLAS analysis [16] provides in Fig. 12 signal strengths for $H \to \gamma\gamma$ separated into ggH, VBF, VH and "top" (ttH+tH) production modes. No correlations are given for the signal strengths, but we can use instead the correlations for the stage-0 STXS provided in Fig. 40a of the ATLAS paper, which should be a close enough match. It turns out, however, that the $\mu$ values rounded to one decimal do not allow to reproduce very well the ATLAS coupling fits for ($C_V$, $C_F$) or ($C_\gamma$, $C_g$). We have therefore extracted the best-fit points and uncertainties from the 1D profile likelihoods, which are provided as Auxiliary Figures 23a–d on the analysis webpage, as[7] $\mu(\text{ggH}, \gamma\gamma) \simeq 0.81^{+0.19}_{-0.18}$, $\mu(\text{VBF}, \gamma\gamma) \simeq 2.04^{+0.61}_{-0.53}$, $\mu(\text{VH}, \gamma\gamma) \simeq 0.66^{+0.89}_{-0.80}$ and $\mu(\text{ttH}, \gamma\gamma) \simeq 0.54^{+0.64}_{-0.55}$ (using a Poisson likelihood). These numbers are consistent with the rounded values in Fig. 12 of [16], but using more digits improves the coupling fits as shown in Fig. 1.

---

[7]In the XML file, we use the exact numbers from the fit to the 1D profile likelihoods.

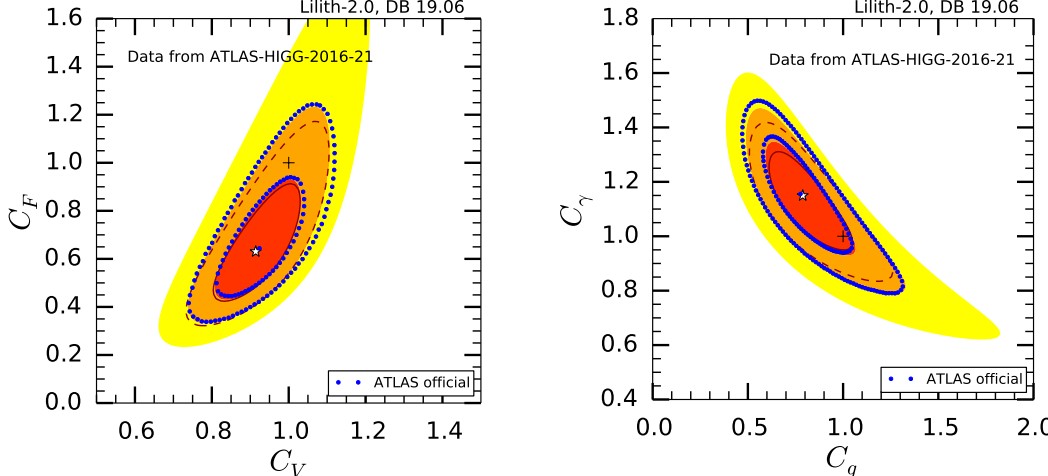

Figure 1: Fit of $C_F$ vs. $C_V$ (left) and $C_\gamma$ vs. $C_g$ (right) for data from the ATLAS $H \to \gamma\gamma$ analysis [16]. The red, orange and yellow filled areas show the 68%, 95% and 99.7% CL regions obtained with `Lilith` using best-fit values and uncertainties for the signal strengths as extracted from Aux. Figs. 23a–d of the ATLAS analysis together with the $4 \times 4$ correlation matrix for the stage-0 STXS. This can be compared to the 68%, 95% CL contours obtained using the rounded values from Fig. 12 of [16] (solid and dashed dark red lines) and to the official 68% and 95% CL contours from ATLAS (blue dots). The best-fit point from `Lilith` (ATLAS) is marked as a white star (blue dot), and the SM as a +.

**$H \to ZZ^* \to 4l$ (HIGG-2016-22):** A similar issue as discussed for $H \to \gamma\gamma$ above arises for $H \to ZZ^*$. In order to reasonably reproduce the $C_F$ vs. $C_V$ fit of ATLAS (Fig. 8b of [21]), we fit the 1D profile likelihoods for $\mu(\text{ggH}, ZZ^*)$ and $\mu(\text{VBF}, ZZ^*)$ shown in Aux. Figs. 7a and 7b of [21] as Poisson distributions. This gives $\mu(\text{ggH}, ZZ^*) \simeq 1.12^{+0.25}_{-0.22}$ and $\mu(\text{VBF}, ZZ^*) \simeq 3.88^{+1.75}_{-1.46}$, which we implement as a bivariate Poisson distribution with correlation $\rho = -0.41$ (from Aux. Fig. 4c of [21]). For the VH and ttH production modes, lacking more information, we convert the given 95% CL limits into $\mu(\text{VH}, ZZ^*) = 0 \pm 1.89$ and $\mu(\text{ttH}, ZZ^*) = 0 \pm 3.83$ using a 2-sided Gaussian (assuming 1-sided limits gives a less good agreement with the ATLAS $C_F$ vs. $C_V$ fit). The validation is shown in Fig. 2 (see also Fig. 14 in Appendix A). This is a case where the variable Gaussian approximation performs less well than the Poisson likelihood.

**$H \to WW^* \to 2l2\nu$ (HIGG-2016-07 and HIGG-2017-14):** Ref. [22] focusses on the measurement of the inclusive ggH and VBF Higgs production cross sections in the $H \to WW^* \to e\nu\mu\nu$ channel. The paper quotes on page 13 signal strengths of $\mu(\text{ggH}, WW) = 1.10^{+0.21}_{-0.20}$ and $\mu(\text{VBF}, WW) = 0.62^{+0.36}_{-0.35}$. We implemented these as a 2D result with a correlation of $\rho = -0.08$ using the variable Gaussian approximation; the correlation was fitted from the $\sigma \times$ BR plot, Fig. 9, of [22]. In addition, Ref. [27] presents the measurement of the $H \to WW^* \to \ell\nu\ell\nu$ $(\ell = e, \mu)$ channel for Higgs boson production in association with a vector boson. Using again the variable Gaussian approximation, we extracted $\mu(\text{WH}, WW) \simeq 2.29^{+1.19}_{-1.01}$ and $\mu(\text{ZH}, WW) \simeq 2.86^{+1.87}_{-1.33}$ with correlation $\rho = -0.08$ from Fig. 8 of that paper. As no other validation material is available, we show in Fig. 3 (top left and top right plots) our reconstruction of the experimental likelihoods in the $\mu(\text{ggH}, WW)$ vs. $\mu(\text{VBF}, WW)$ and $\mu(\text{ZH}, WW)$ vs. $\mu(\text{WH}, WW)$ planes, comparing respectively to the rescaled

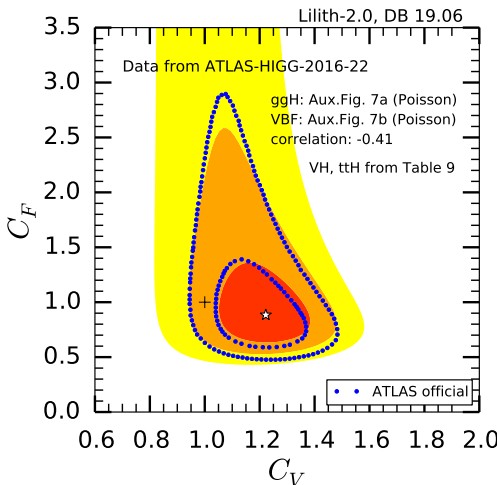

Figure 2: Fit of $C_F$ vs. $C_V$ for data from the ATLAS $H \to ZZ^*$ analysis, using $\mu(\text{ggH}, ZZ^*)$ and $\mu(\text{VBF}, ZZ^*)$ as fitted from Aux. Figs. 7a and 7b of [21]; the ggH vs. VBF likelihood is then approximated as a bivariate Poissonian with correlation $-0.41$ (see text for more details). The 68%, 95% and 99.7% CL regions obtained with `Lilith` are shown as red, orange and yellow areas, and compared to the 68% and 95% CL contours from ATLAS (in blue). The best-fit point from `Lilith` is marked as a white star and the SM as a +.

contours of Fig. 9 of [22] and Fig. 8 of [27].[8]

$H \to \tau\tau$ **(HIGG-2017-07):** This ATLAS cross section measurement in the $H \to \tau\tau$ channel [23] provides as Aux. Fig. 5 the 68% and 95% CL contours in the $\mu(\text{ggH}, \tau\tau)$ vs. $\mu(\text{VBF}, \tau\tau)$ plane. A fit of a bivariate variable Gaussian to the 95% CL contour in this plot gives $\mu(\text{ggH}, \tau\tau) \simeq 1.0^{+0.72}_{-0.59}$ and $\mu(\text{VBF}, WW) \simeq 1.20^{+0.62}_{-0.56}$ with $\rho \simeq -0.45$, which are the values implemented in the database. As for $H \to WW$ above, no coupling fits are available which could be used for validation. We therefore show in Fig. 3 (bottom plot) our reconstruction of the experimental likelihood in the $\mu(\text{ggH}, \tau\tau)$ vs. $\mu(\text{VBF}, \tau\tau)$ plane. Note that a fit to the 68% CL contour of ATLAS gives a less good result.

$H \to \mu\mu$ **(HIGG-2016-10):** In Ref. [24], ATLAS reports a measured overall signal strength of $\mu(H \to \mu\mu) = -0.1 \pm 1.5$, from which a 95% CL limit of 3.0 is computed using the $\text{CL}_s$ prescription. For consistency with the 95% CL limit, that is to avoid being overconstraining, we implement this as $\mu(H \to \mu\mu) = 0 \pm 1.53$ in the `Lilith` database. The relative contributions of ggH (90%), VBF (7%) and VH (3%) production are estimated from Aux. Table 3 on the analysis' webpage, using the sum of all eight orthogonal categories.

$H \to b\bar{b}$ **(HIGG-2016-29 and HIGG-2016-30):** For the $H \to b\bar{b}$ decay mode, ATLAS gives $\mu(\text{ZH}, b\bar{b}) = 1.12^{+0.50}_{-0.45}$, $\mu(\text{WH}, b\bar{b}) = 1.35^{+0.68}_{-0.59}$ [28] and $\mu(\text{VBF}, b\bar{b}) = 3.0^{+1.7}_{-1.6}$ [25]. No correlation data is available, so we implemented each of these as a 1D result; a Poisson likelihood is assumed per default but can easily be changed to a variable Gaussian if the user wishes to do so.

---

[8]Here and in the following, whenever the experimental paper gives only measured cross sections but no signal strengths, we use the recommended SM cross section values from the LHC Higgs Cross Section Working Group [15] for appropriate rescaling.

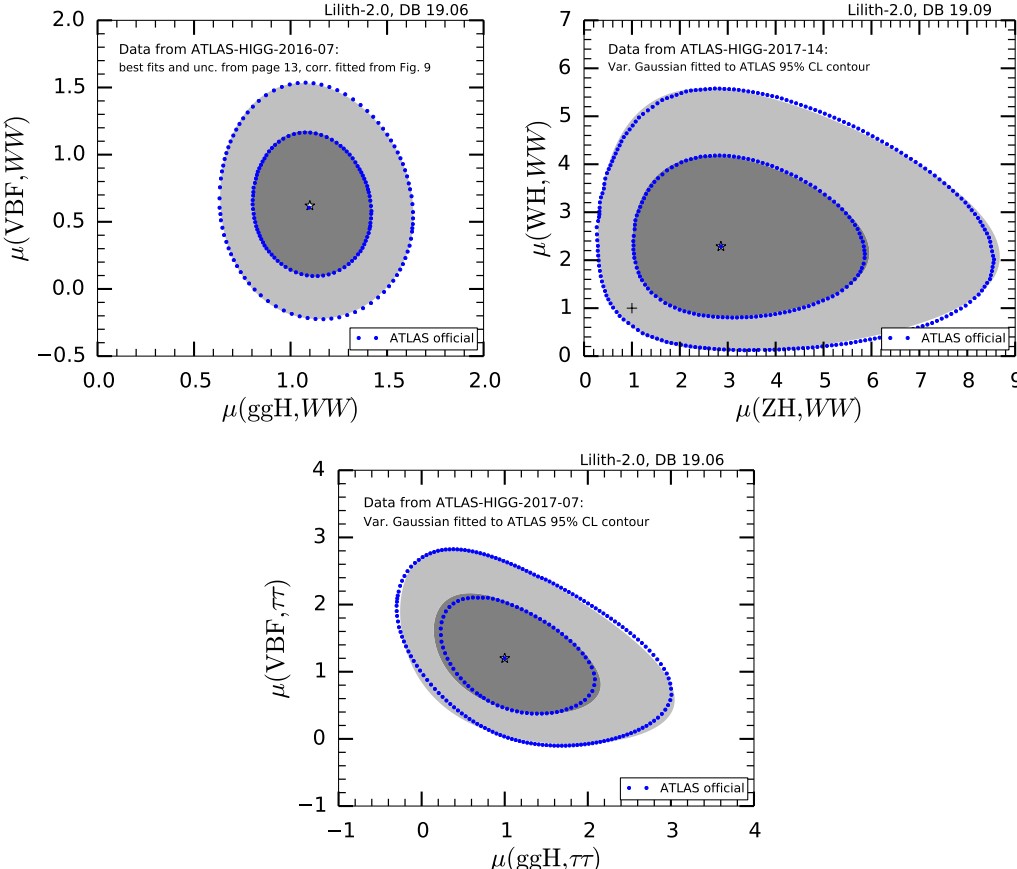

Figure 3: Reconstruction of the experimental likelihood as 2D variable Gaussian; top panels for the $H \to WW$ channel from [22, 27], bottom panel for the $H \to \tau\tau$ channel from [23]. The 68% and 95% CL regions obtained with `Lilith` are shown in dark and light gray, respectively, and compared to the 68% and 95% CL contours from ATLAS (in blue). The best-fit points from `Lilith` and ATLAS are marked as white stars and blue dots, respectively.

**ttH production (HIGG-2017-02):** The ATLAS paper [30], reporting evidence for $t\bar{t}H$ production, provides in Fig. 16 the signal strength results broken down into $H \to \gamma\gamma$, $VV$ ($= ZZ^* + WW^*$), $\tau\tau$ and $b\bar{b}$ decay modes from a combined analysis of all $t\bar{t}H$ searches. Correlations are not given explicitly but can be estimated from Figs. 17a and 17b in [30] as $\rho(b\bar{b}, VV) \simeq 0.04$ for the correlation between the $H \to b\bar{b}$ and $H \to VV$ decay modes and $\rho(\tau\tau, VV) \simeq -0.35$ for that between the $H \to \tau\tau$ and $H \to VV$ decay modes. For validation, we compare in Fig. 4 the $C_F$ vs. $C_V$ fit from the implementation in `Lilith` to the official one from [30].

A few comments are in order here. First, the measurement of $\mu(ttH, \gamma\gamma)$ given in [30] actually comes from [16] (HIGG-2016-21, see above) and is also included in the HIGG-2016-21 XML file; to avoid overlap when using both the HIGG-2016-21 and HIGG-2017-02 datasets, we provide a 3D XML file for the latter which includes only the $VV$, $\tau\tau$ and $b\bar{b}$, but not the $\gamma\gamma$ decay mode. The important point however is that the value given by ATLAS is not $\mu(ttH, \gamma\gamma)$ but $\mu(ttH + tH, \gamma\gamma)$.[9] This makes a big difference in the validation plot. Second, the individual measurement [31] gives $\mu(ttH, b\bar{b})$ to two decimals $(0.84^{+0.64}_{-0.61})$ instead just one $(0.8 \pm 0.6)$ in [30]. Again this makes a visible difference in Fig. 4, improving the quality of the fit, so we

---

[9]Lacking more precise information, ttH and tH production are combined according to Eq. (19).

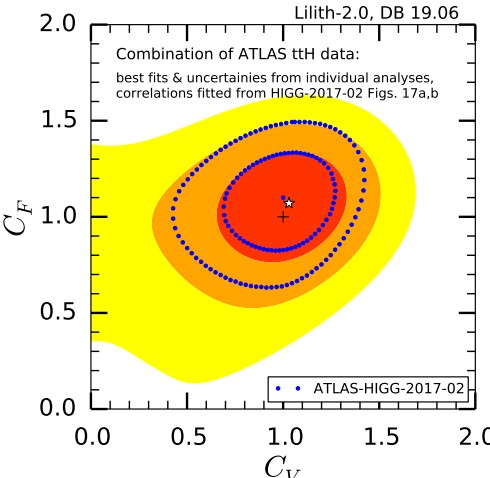

Figure 4: Fit of $C_F$ vs. $C_V$ from a combination of the ATLAS ttH measurements (see text for details). The 68%, 95% and 99.7% CL regions obtained with `Lilith` are shown as red, orange and yellow areas, and compared to the 68%, 95% CL contours from ATLAS (in blue). The best-fit point from `Lilith` (ATLAS) is marked as a white star (blue dot), and the SM as a +.

use the more precise numbers from [31]. The relevance of these points, and of the fitted correlations, is illustrated in Fig. 15 in Appendix A. Third, for $\mu(t\bar{t}H, VV)$ the contribution from $H \to WW^*$ should dominate, but the concrete weights of the $ZZ^*$ and $WW^*$ decay modes are not given in [30]. (We use 95% for $WW^*$ and 5% for $ZZ^*$ as a rough estimate.) This is not a problem as long as $C_Z = C_W \equiv C_V$, but one should not use the HIGG-2017-02 XML file for any other case.

**$H \to$ invisible (HIGG-2016-28 and HIGG-2018-54):** Results from the search for invisibly decaying Higgs bosons produced in association with a $Z$ boson are presented in [29]. A 95% CL upper limit of BR($H \to$ inv.) < 0.67 is set for $m_H = 125$ GeV assuming the SM $ZH$ production cross section. In the `Lilith` database, we use a likelihood grid as function BR($H \to$ inv.) extracted from Aux. Fig. 1c on the analysis' webpage. The combination of Run 2 searches for invisible Higgs boson decays in [26] tightens BR($H \to$ inv.) < 0.38 at 95% CL; here, we use the likelihood grid for VBF production of invisibly decaying Higgs bosons extracted from Fig. 1a.

## 5.2 CMS Run 2 results for 36 fb$^{-1}$

The CMS Run 2 results included in this release are summarised in Table 4 and explained in more detail below.

**Combined measurements (HIG-17-031):** CMS presented in [12] a combination of the individual measurements for the $H \to \gamma\gamma$ [34], $ZZ$ [35], $WW$ [36], $\tau\tau$ [37], $b\bar{b}$ [38,39] and $\mu\mu$ [40] decay modes as well as the $t\bar{t}H$ analyses [41–43]. We use the best fit values and uncertainties for the signal strengths for each production and decay mode combination presented in Table 3 of [12] together with the $24 \times 24$ correlation matrix provided as "Additional Figure 1" on the analysis webpage. Implemented as a variable Gaussian likelihood, this allows to reproduce well the coupling fits of the CMS paper as shown in Figs. 5 and 6.

**$VH, H \to \tau\tau$ (HIG-18-007):** The above data from [12] is supplemented by the results for the $\tau\tau$ decay mode from the $WH$ and $ZH$ targeted analysis [33]. These are implemented in

Table 4: Overview of CMS Run 2 results included in this release. Note that we use the full $24 \times 24$ correlation matrix for the signal strengths for each production and decay mode combination provided in [12].

| mode | $\gamma\gamma$ | $ZZ^*$ | $WW^*$ | $\tau\tau$ | $b\bar{b}$ | $\mu\mu$ | inv. |
|------|------|------|------|------|------|------|------|
| ggH | [12] | [12] | [12] | [12] | [12] | [12] | [32] |
| VBF | [12] | [12] | [12] | [12] | – | [12] | [32] |
| WH | [12] | [12] | [12] | [33] | [12] | – | [32] |
| ZH | [12] | [12] | [12] | [33] | [12] | – | [32] |
| ttH | [12] | [12] | [12] | [12] | [12] | – | – |

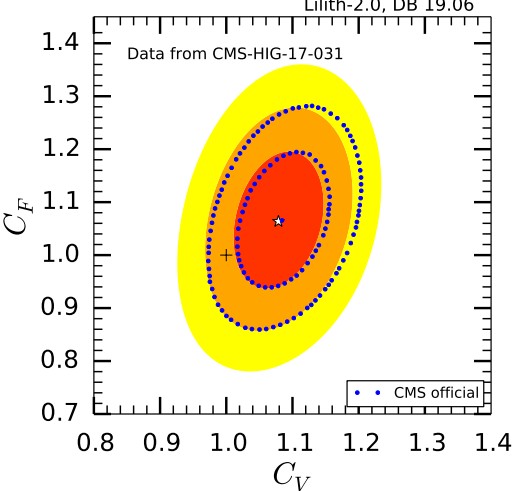

Figure 5: Fit of $C_F$ vs. $C_V$ using best-fit values and uncertainties for the signal strengths for each production (ggH, VBF, WH, ZH, ttH) and decay ($\gamma\gamma$, $ZZ$, $WW$, $\tau\tau$, $b\bar{b}$, $\mu\mu$) mode combination together with the $24 \times 24$ correlation matrix from the CMS combination paper [12]. The $1\sigma$, $2\sigma$ and $3\sigma$ regions obtained with `Lilith` are shown as red, orange and yellow areas, and compared to the $1\sigma$ and $2\sigma$ contours from CMS (blue dots). The best-fit point from `Lilith` (CMS) is marked as a white star (blue dot), the SM as a +.

the form of 1D intervals for $\mu(ZH, \tau\tau)$ and $\mu(WH, \tau\tau)$ taken from Fig. 6 of [33].

**$H \to$ invisible (HIG-17-023)**: In [32], CMS performed a search for invisible decays of a Higgs boson produced through vector boson fusion, setting a limit of BR($H \to$ inv.) $< 0.33$ at 95% CL. We use the profile likelihood ratios for the qqH-, Z(ll)H-, V(qq')H- and ggH-tag categories extracted from their Fig. 8b together with the relative contributions from the different Higgs production mechanisms given in Table 6 of that paper. This assumes that the relative signal contributions stay roughly the same as for SM production cross sections. For validation, we reproduce in Fig. 7 the $C_g$ vs. $C_\gamma$ fit of [12], where the branching ratios of invisible and undetected decays are treated as free parameters.[10]

---

[10]The profiling in Fig. 7 was done with `Minuit`. Since `Minuit` does not allow conditional limits, in this case BR($H \to$ inv.) + BR($H \to$ undetected) $< 1$, we demanded that both BR($H \to$ inv.) and BR($H \to$ undetected) be less than 50%.

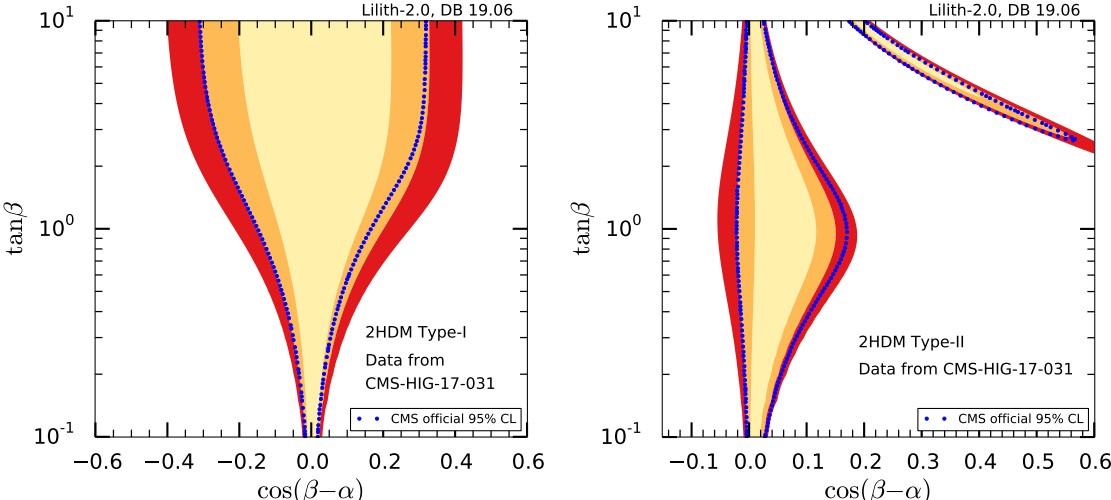

Figure 6: Fit of $\tan\beta$ vs. $\cos(\beta-\alpha)$ for the 2HDMs of Type I (left) and Type II (right) using the data from the combined CMS measurement [12]. The beige, orange and red filled areas show the 68%, 95% and 99.7% CL regions obtained with Lilith, while the blue dots mark the 95% CL contours from CMS.

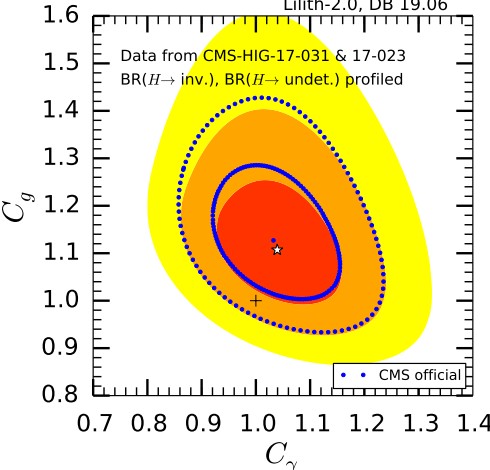

Figure 7: Fit of $C_g$ vs. $C_\gamma$ using the data from the combined CMS measurement [12] and the search for invisible decays of a Higgs boson [32]. The branching ratios of invisible and undetected decays are treated as free parameters in the fit. The $1\sigma$, $2\sigma$ and $3\sigma$ regions obtained with Lilith are shown as red, orange and yellow areas, and compared to the $1\sigma$ and $2\sigma$ contours from CMS (in blue). The best-fit point from Lilith (CMS) is marked as a white star (blue dot), and the SM as a +.

## 6 Status of Higgs coupling fits

In this section we give a brief overview of the current status of Higgs coupling fits. We remind the reader that these fits rely on the specific assumptions mentioned in the Introduction and do not represent general, model-independent statements on "the Higgs couplings". Most importantly, new physics effects are assumed to follow the same Lorentz structure as the SM Higgs couplings, such that new physics contributions to the Higgs production and decay processes can be parameterised in terms of reduced coupling factors and some coefficients depending

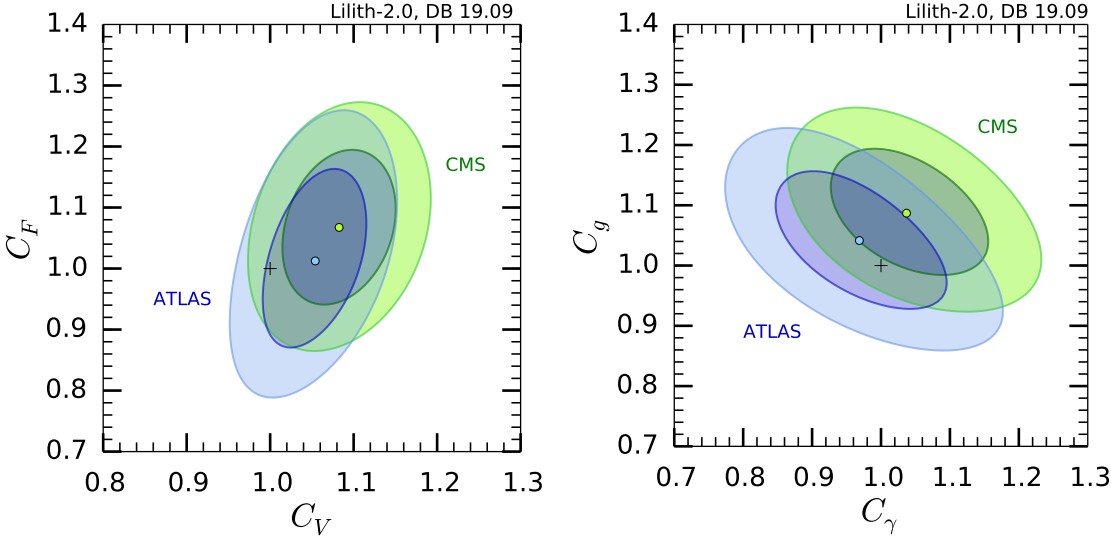

Figure 8: Fit of $C_F$ vs. $C_V$ (left) and $C_g$ vs. $C_\gamma$ (right) using the Run 2 dataset of the current database version, DB 19.09. The 68% and 95% CL regions for the combined ATLAS results are shown in blue, those for CMS in green.

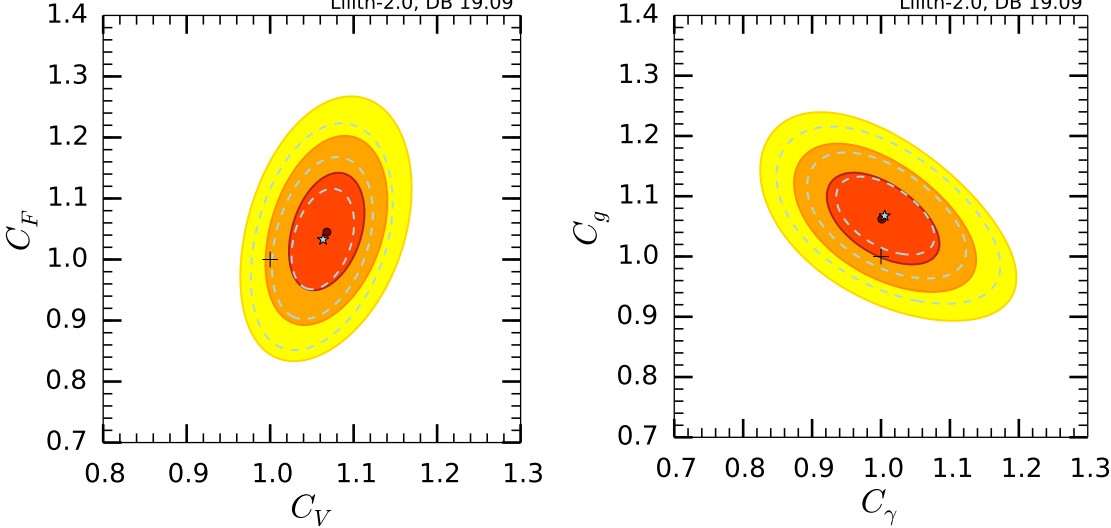

Figure 9: Fit of $C_F$ vs. $C_V$ (left) and $C_g$ vs. $C_\gamma$ (right) from a combination of the ATLAS and CMS Run 2 results in DB 19.09; the 68%, 95% and 99.7% CL regions are shown as red, orange and yellow areas, respectively. In addition, the light-blue, dashed contours indicate the 68%, 95% and 99.7% CL regions when combining the Run 2 and Run 1 data. The best-fit points for Run 2 (Run 2+Run 1) data are marked as red dots (light-blue stars), the SM as black +.

only on SM predictions, as seen e.g. from Eqs. (20)–(23) in Section 4. Details about the accuracy of the SM predictions are also provided in Section 4. In all that follows, we use $m_H = 125.09$ GeV. For other global fits to Run 2 Higgs results see, e.g., [44–47].

We begin by showing in Fig. 8 fits of $C_F$ vs. $C_V$ (left panel) and $C_g$ vs. $C_\gamma$ (right panel) using either the ATLAS (in blue) or the CMS (in green) Run 2 results in the current `Lilith` database, DB 19.09. As can be seen, the two experiments agree at the level of about $1\sigma$, the ATLAS results being slightly closer to the SM (marked as a black + in all plots).

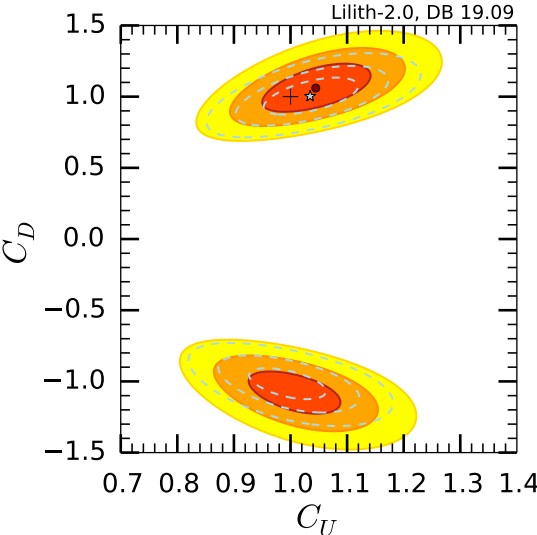

Figure 10: As Fig. 9 but for a fit of $C_D$ vs. $C_U$ with $C_V$ profiled over.

The situation when combining the results from both experiments is shown in Fig. 9. Using the Run 2 (Run 2 + Run 1) results of DB 19.09,[11] we find with the help of `Minuit`

$$C_F = 1.045^{+0.064}_{-0.063} \ (1.033^{+0.055}_{-0.054}), \quad C_V = 1.068 \pm 0.030 \ (1.063 \pm 0.025), \tag{24}$$

with a correlation of 0.33 (0.33). This assumes that contributions from new particles to the loop-induced couplings to gluons and photons as well as invisible or undetected decays are absent. Comparing the SM to the $(C_F, C_V)$ best fit gives $-2\log(L_{SM}/L_{max}^{C_F,C_V}) = 5.13$ (6.47), corresponding to a $p$-value of 0.16 (0.09) based on Run 2 (Run 2 + Run 1) results.

Taking instead $C_g$ and $C_\gamma$ as free parameters with $C_F = C_V = 1$ (still assuming that invisible or undetected decays are absent), gives

$$C_g = 1.062^{+0.051}_{-0.050} \ (1.068 \pm 0.043), \quad C_\gamma = 1.001^{+0.055}_{-0.053} \ (1.005^{+0.048}_{-0.047}), \tag{25}$$

with correlation $-0.52$ $(-0.51)$ from Run 2 (combining Run 2 and Run 1) results; here the SM point has a $p$-value of 0.34 (0.16).

It is also interesting to consider the couplings to up-type and down-type fermions as independent parameters. In this case, we find

| $C_D > 0$: Run 2 | (Run 2+Run 1) | $C_D < 0$: Run 2 | (Run 2+Run 1) |
|---|---|---|---|
| $C_U = 1.04 \pm 0.06$ | $(1.03^{+0.06}_{-0.05})$ | $C_U = 1.01 \pm 0.06$ | $(0.99^{+0.06}_{-0.05})$ |
| $C_D = 1.06 \pm 0.11$ | $(1.00 \pm 0.09)$ | $C_D = -1.08^{+0.11}_{-0.12}$ | $(-1.01 \pm 0.09)$ |
| $C_V = 1.08 \pm 0.06$ | $(1.05 \pm 0.04)$ | $C_V = 1.08 \pm 0.06$ | $(1.05 \pm 0.04)$ |

(26)

where we fitted separately for the two possible solutions of same-sign or opposite-sign $C_D$ with respect to $C_U, C_V > 0$. With $-2\log L_{max}^{C_D>0} = 43.25$ (48.11) compared to $-2\log L_{max}^{C_D<0} = 43.86$ (48.83), neither solution is clearly preferred by the data. Contours of constant CL in the $C_D$ vs. $C_U$ plane with $C_V$ profiled over can be seen in Fig. 10.

The $(C_F, C_V)$ and $(C_U, C_D, C_V)$ fits above have their correspondence in the 2HDM of Type I and Type II, albeit $C_V$ is restricted to $C_V \leq 1$ in 2HDMs (and generally in models with only

---

[11]For Run 1, we use the results from the official ATLAS+CMS combination [10] available in DB 19.09, plus the individual ATLAS and CMS $H \to$ inv. results. We also note here that for the SM, we get $-2\log L_{SM} = 48.40$ using the Run 2 results in DB 19.09 (53 measurements) and $-2\log L_{SM} = 54.78$ from the combination of Run 2 and Run 1 results (66 measurements).

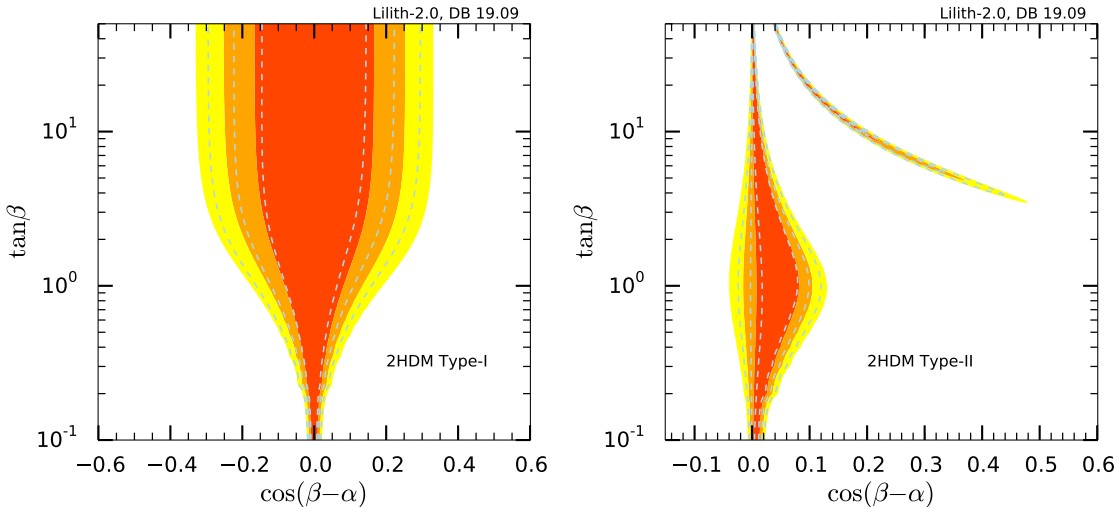

Figure 11: Fits of $\tan\beta$ vs. $\cos(\beta - \alpha)$ for the 2HDM of Type I (left) and of Type II (right) from a combination of the ATLAS and CMS Run 2 results in DB 19.09. The red, orange and yellow areas are the 68%, 95% and 99.7% CL regions, respectively. In addition, the light-blue, dashed contours indicate the 68%, 95% and 99.7% CL regions when combining the Run 2 and Run 1 data. Loop contributions from charged Higgs bosons are neglected and decays into non-SM particles (such as $h \to AA$) assumed to be absent.

Higgs doublets and singlets). The couplings of the lighter scalar $h$ are $C_F = \cos\alpha/\sin\beta$ in Type I, and $C_U = \cos\alpha/\sin\beta$ and $C_D = -\sin\alpha/\cos\beta$ in Type II; $C_V = \sin(\beta - \alpha)$ in both models. The fit results in the $\tan\beta$ vs. $\cos(\beta - \alpha)$ plane are shown in Fig. 11. Note that for Type II the banana-shaped second branch corresponds to the "opposite-sign" solution for the bottom Yukawa coupling [48].

Before concluding, let us turn to invisible Higgs decays. Figure 12 (left) shows the 1D profile likelihood of BR($H \to$ inv.) for two cases, SM couplings (in red) and $C_F$ and $C_V$ as free parameters (in blue). We find that the Run 2 results in DB 19.09 constrain BR($H \to$ inv.) $\lesssim 5\%$ at 95% CL for the SM-like case,[12] and to BR($H \to$ inv.) $\lesssim 16\%$ when $C_F$ and $C_V$ are treated as free parameters; the case of free $C_U$, $C_D$, $C_V$ is not shown but gives the same result. For Run 2+Run 1 results, these values tighten to BR($H \to$ inv.) $\lesssim 4\%$, 15%, 15% for SM couplings, free $C_F$, $C_V$, and free $C_U$, $C_D$, $C_V$, respectively. For completeness, Fig. 12 (right) shows the 68%, 95% and 99.7% CL regions from a 2D fit of BR($H \to$ inv.) vs. $C_V$ with $C_F$ profiled over.

---

[12]This strong bound is in fact primarily driven by the signal strength measurements in the SM final states, as invisible decays reduce the branching ratios to (visible) SM particles [49].

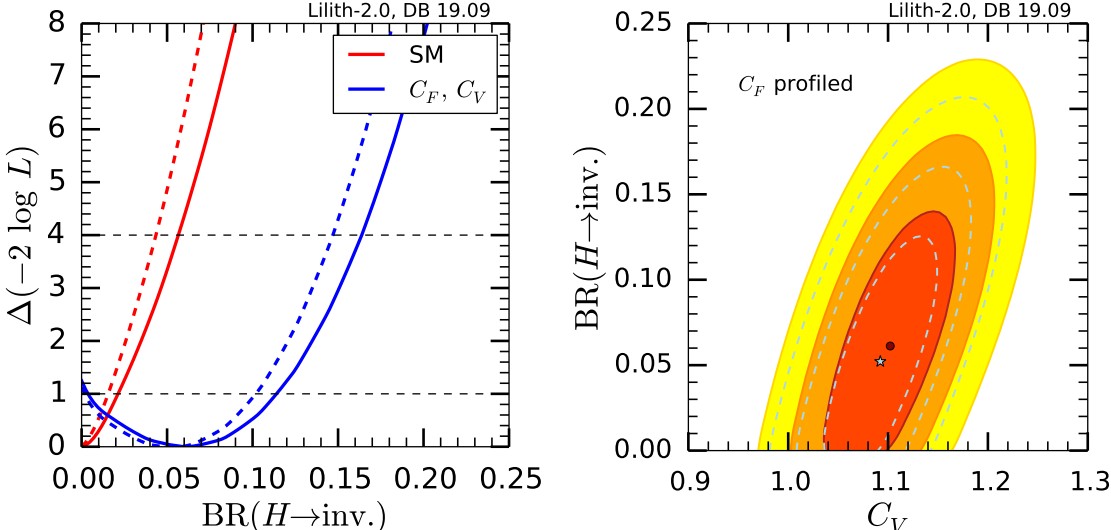

Figure 12: Status of invisible Higgs decays. Left: 1D profile likelihood of BR($H \to$ inv.), in red for SM ($C_F = C_V = 1$) couplings, in blue for $C_F$, $C_V$ as free parameters; full lines are for Run 2, while dashed lines are for Run 2 + Run 1 results in DB 19.09. Right: 2D fit of BR($H \to$ inv.) vs. $C_V$ with $C_F$ profiled over; same style as in Fig. 9.

## 7   Conclusions

We presented `Lilith-2.0`, a light and easy-to-use Python tool for constraining new physics from signal strength measurements of the 125 GeV Higgs boson. The main novelties include

- a better treatment of asymmetric uncertainties through the use of variable Gaussian or Poisson likelihoods where appropriate;

- the use of multi-dimensional correlations whenever available;

- a new database (DB 19.09) including the published ATLAS and CMS Run 2 Higgs results for 36 fb$^{-1}$.

We provided detailed validations of the results included in DB 19.09 and discussed the consequences of the available Run 2 results for fits of reduced Higgs couplings, 2HDMs of Type I and Type II, and invisible Higgs decays. Our analysis shows that the ATLAS and CMS results well agree with each other. The data is perfectly compatible with the SM, putting very tight constraints on any deviations. Indeed, our combination of the ATLAS and CMS results in global fits of ($C_F$, $C_V$), ($C_g$, $C_\gamma$) or ($C_U$, $C_D$, $C_V$) leads to a determination of these couplings to better than 10%. In particular, the uncertainty on $C_V$ shrinks to about 3–4%, and we observe a slight preference for $C_V > 1$. In the context of 2HDMs, where $C_V \leq 1$, this forces one even deeper into the alignment limit [50, 51]. Finally, the global fit also tightly constrains invisible Higgs decays — for SM-like couplings, to BR($H \to$ inv.) $\lesssim$ 5% at 95% CL.

`Lilith-2.0` with its latest database is publicly available at

http://lpsc.in2p3.fr/projects-th/lilith/

or directly on GitHub at https://github.com/sabinekraml/Lilith-2 and ready to be used to constrain a wide class of new physics scenarios. `Lilith` is also interfaced from `micrOMEGAs` [52] (v4.3 or higher). Readers who already have `Lilith-1.1` in their `micrOMEGAs` installation can simply replace it with the new version `2.0`.

Given the high interest and ease of use of the signal strength framework, we kindly ask the experimental collaborations to continue to provide detailed signal strength results for the pure Higgs production×decay modes, including their correlations. The CMS combination paper [12] is an example of good practice in this respect. We want to stress, however, that when results are given for a combination of production and/or decay modes, it is important that the relative contributions be given and all assumptions be clearly spelled out. This is currently not the case in all publications. Moreover, as we have shown, it is crucial for a good usage of the experimental results, that the numbers quoted in tables and plots be precise enough. Such issues made the validation of some of the results in DB 19.09 seriously difficult, and we dearly hope that this will improve in the future.

Last but not least, we want to note that extracting results by digitizing curves from a plot, or typing the numbers for a large correlation matrix which is only available as an image, is painful, prone to errors, and should not be necessary in the modern information age. *We therefore implore that all results be made available in digitized form, be it on HEPData or on the collaboration twiki page, by the time an analysis is published.*

# Acknowledgements

S.K. thanks R. Schöfbeck, W. Waltenberger and N. Wardle for helpful discussions. This work was supported in part by the IN2P3 theory project "LHC-itools: methods and tools for the interpretation of the LHC Run 2 results for new physics". D.T.N and L.D.N. are funded by the Vietnam National Foundation for Science and Technology Development (NAFOSTED) under grant number 103.01-2017.78. D.T.N. thanks the LPSC Grenoble for hospitality and financial support for a research visit within the LHC-itools project. T.Q.L. acknowledges the hospitality and financial support of the ICISE Quy Nhon.

# A  Comparison with alternative implementations of the experimental results

To illustrate the importance of various improvements discussed throughout the paper, we here show versions of validation plots for alternative implementations of the experimental results. These can be compared with the corresponding validation plots for the official release in Section 5.

To start with, we show in Fig. 13 fits of $C_F$ vs. $C_V$, where we used the ordinary Gaussian approximation (i.e. with fixed width) instead of the variable Gaussian. The plot on the left is for the ATLAS $H \rightarrow \gamma\gamma$ [16] data, which has a $4 \times 4$ correlation matrix for the ggH, VBF, VH and ttH+tH production modes. The plot on the right is for the CMS combination [12], which has a $24 \times 24$ correlation matrix. It is obvious that the fixed-width Gaussian does not correctly approximate the true likelihood. The variable Gaussian implementation, on the other hand, gives a satisfying result.

In some cases with large asymmetries, a Poisson distribution is more appropriate than a variable Gaussian. This is the case for the ATLAS $H \rightarrow ZZ^* \rightarrow 4l$ analysis [21] as shown in Fig. 14. Note, however, that although the Poisson form allows to better reproduce the 68% CL contour of ATLAS than the variable Gaussian, the 95% CL contour is still quite off. This is much improved by using the 1D profile likelihoods for $\mu(\text{ggH}, ZZ^*)$ and $\mu(\text{VBF}, ZZ^*)$ from Aux. Figs. 7a,b of [21] (also parametrised as Poisson likelihoods) instead of the best-fit and uncertainty values from Table 9 of [21], see Fig. 2.

Finally, Fig. 15 details the steps we made to achieve a good implementation and validation

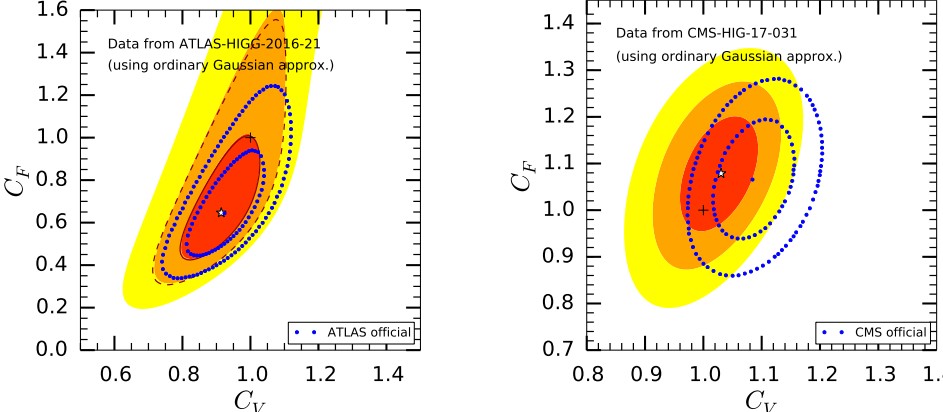

Figure 13: Fits of $C_F$ vs. $C_V$ for the data from ATLAS-HIGG-2016-21 (left) and CMS-HIG-17-031 (right) using `type="n"` (ordinary Gaussian) instead of `type="vn"` (variable Gaussian) in the database XML file. To be compared with the respective plots in Figs. 1 and 5.

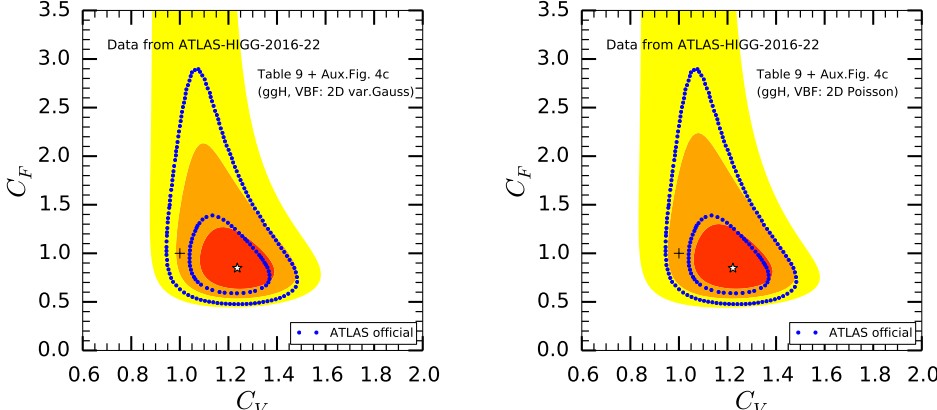

Figure 14: Fits of $C_F$ vs. $C_V$ for the data from ATLAS-HIGG-2016-22 ($H \rightarrow ZZ^*$), using $\mu(\text{ggH}, ZZ^*) = 1.11^{+0.23}_{-0.21}$ and $\mu(\text{VBF}, ZZ^*) = 4.0^{+1.75}_{-1.46}$ from Table 9 of [21] with correlation $\rho = -0.41$ from Aux. Fig. 4c on the analysis twiki page. On the left, the likelihood of $\mu(\text{ggH}, ZZ^*)$ vs. $\mu(\text{VBF}, ZZ^*)$ is parametrised as a 2D variable Gaussian, on the right as a 2D Poisson distribution. To be compared with Fig. 2, where $\mu(\text{ggH}, ZZ^*) \simeq 1.12^{+0.25}_{-0.22}$ and $\mu(\text{VBF}, ZZ^*) \simeq 3.88^{+1.75}_{-1.46}$, fitted from Aux. Figs. 7a,b of [21], are used to construct a 2D Poisson likelihood. VH and ttH production are treated as in Fig. 2.

of the ATLAS ttH combination. The labels "best fits & uncertainties as given in HIGG-2017-02 (v1)" and "... (v2)" refer to identifying the measurement in the $\gamma\gamma$ final state as $\mu(\text{ttH}, \gamma\gamma)$ or $\mu(\text{ttH} + \text{tH}, \gamma\gamma)$. See also the discussion related to Fig. 4 in Section 5.1.

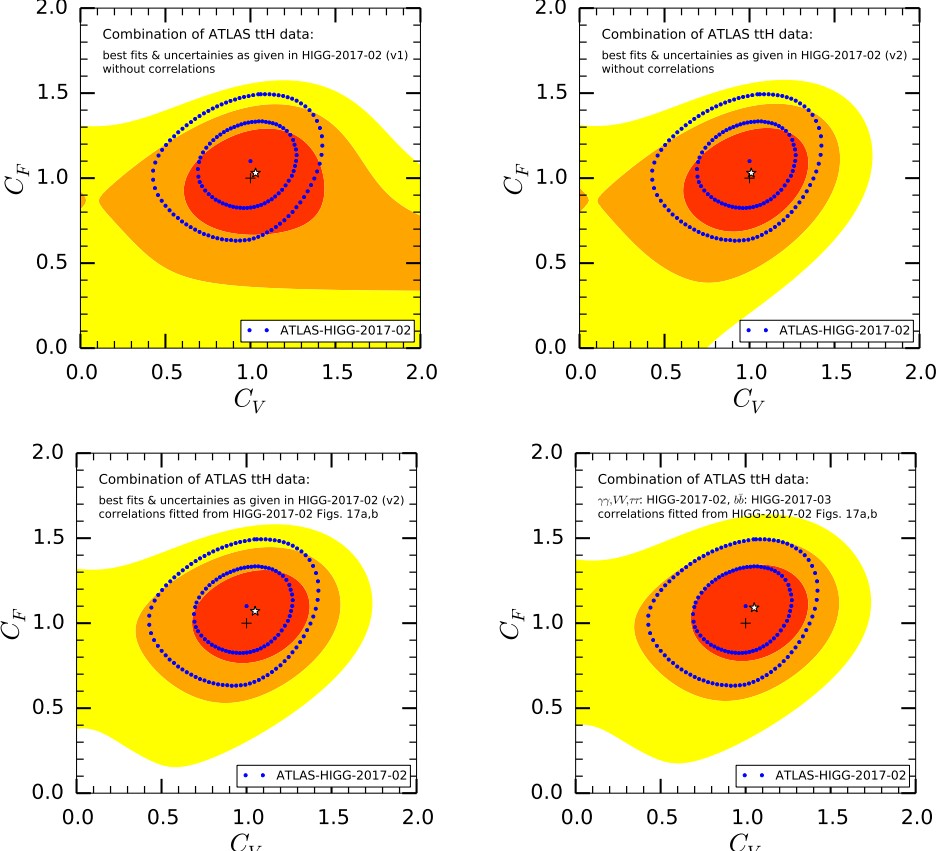

Figure 15: Evolution of the validation for the ATLAS ttH combination. Top left: $\mu(\text{ttH}, Y)$ for $Y = \tau\tau$, $\gamma\gamma$, $b\bar{b}$, $VV$ as quoted in Fig. 16 of ATLAS-HIGG-2017-02 [30]. Top right: same as on the left but including tH production for the $\gamma\gamma$ channel according to the second bullet point on p. 37 of [30]. Bottom left: adding the correlations fitted from Figs. 17a,b of [30]. Bottom right: using $\mu(\text{ttH}, b\bar{b})$ from [31] instead of the value from [30]. The bottom right plot is already very close to the final version shown in Fig. 4. The only difference is that for the latter $\mu(\text{ttH} + \text{tH}, \gamma\gamma)$ fitted from the 1D profile likelihood in [16] was used.

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
