# Peer review of "Constraining new physics from Higgs measurements with Lilith: update to LHC Run 2 results"

_SciPost Physics, doi:SciPost Phys. 7, 052 (2019)_

## Round 1 · Referee Report · Anonymous · 2019-8-29

Strengths

1- The provided explanations of the updates of Lilith and how to run them are very detailed and clear.
2- The authors are very explicit about where they took their data (and correlations) from. This makes it very easy to follow and cross check their analyses.
3- The authors carefully validate each of the included data sets, making use not only of the experimental papers but also of auxiliary material online.

Weaknesses

1- Some of the ATLAS analyses used in the combination are avaiblable for a data set with greater luminosity already.
2- Moreover, there are some ATLAS analyses for processes which are not included in the combination at all (see requested changes).

Report

The authors provide a clear and detailed analysis of Higgs coupling strength measurements of LHC Run II data with their Python tool Lilith. With respect to the previous version of Lilith, the authors have updated the statistical analysis framework. It now allows for the inclusion of likelihood contributions as two-sided Gaussians or Poissonians. Moreover, the authors include new Higgs production channels (ggZH and tH production).

The manuscript is clear and well written. I recommend it for publication once the issues below are addressed.

Requested changes

1- On p.7 the authors discuss of the implementation of a variable Gaussian and raise the potential issue of singularities of the log-likelihood function. It is not clear from this discussion how the authors deal with these singularities in practice, what the region of validity of the variable Gaussian implementation is, and why using an ordinary two-sided Gaussian instead (outside of the region of validity) is legitimate in the extremely asymmetrical case $\sigma^{-} \rightarrow 0$. The authors might want to comment on this.

2- Some of the used analyses have been updated for a larger data set already (see weaknesses). The authors could consider updating the analyses available for a higher luminosity, i.e. tth 1806.00425 ($H\rightarrow \gamma \gamma$, $H\rightarrow ZZ$); and VH, $H\rightarrow bb$ 1808.08238.

3- The authors could consider adding a number of potentially relevant Higgs analyses.
* $H\rightarrow$inv: ATLAS WBF analysis 1809.06682. The analysis sets a limit of 37% for $H\rightarrow$inv. This is much better than the limit of 67% obtained in the ZH analysis and comparable to the limit of the CMS WBF analysis of 33%. Including the analysis might have a significant impact on the limit obtained on BR(H->inv).
* $H\rightarrow \mu \mu$: ATLAS 1705.04582; This is the only ATLAS analysis for the mu mu final state. The upper limit on BR($H \rightarrow \mu \mu$) obtained in this analysis is exactly the same as in the CMS analysis (which was included), namely 3.0 time its SM value. It might be worth including the analysis.
* VH, $H\rightarrow WW$: ATLAS 1903.10052; This is the only analysis for this production and decay channel. However, I do agree that $H\rightarrow WW$ is better constrained from ggF and that VH is better constrained from $H\rightarrow bb$, so the impact of this analysis might not be significant in a global fit.

4- Most importantly, the limit on the invisible Higgs branching is interesting, but not obvious. In Fig.12 and the corresponding text on p.18 the authors find a combined 95% CL limit on BR($H\rightarrow$inv) of <= 5% for the SM case. However, the included ATLAS and CMS analyses quote limits of 33% (CMS) and 67% (ATLAS). It is not clear how these limits combine to a 5% limit. The authors should comment on how they obtained this limit and how it compares to the results of the experimental papers.

---

## Round 1 · Referee Report · Anonymous · 2019-9-13

Strengths

1- The manuscript is well written, and the presentation is organized in a clear way.
2- The database of signal strengths included in the package has been systematically validated, as documented in the paper. Complete references to the ATLAS and CMS papers (or to the web versions of the papers, where needed) are provided for each database entry.

Weaknesses

1- The presentation of the modified couplings framework included in the code, and of the fits that are shown as an example, could be slightly improved by stating explicitly the underlying assumptions.

Report

The authors present a major update of the publicly available code Lilith, a tool that implements an approximate global likelihood for the Higgs signal strength measurements, in a form that is suited for comparison against the predictions of extensions of the SM. The manuscript describes the refined statistical approach now employed, that helps to better deal with asymmetric uncertainties, and other improvements of the code. Furthermore, new LHC Run 2 data corresponding to an integrated luminosity of 36/fb has been added, including new production channels as ggZH, tH, bbH. As an example of application, the authors show an updated "kappa-framework" interpretation of the data.

The presentation of the new features in the code is clear and well organized. The validation of the new Lilith dataset is systematically documented. The "kappa-framework" fit that is shown, despite capturing only selected classes of BSM models (being less general than an EFT interpretation, for instance) represents a useful example to illustrate the functionality of the code.

I recommend the manuscript for publication once a few minor points (see below) are addressed.

Requested changes

1- On page 2 the authors describe the assumptions and the scope of the signal strength framework, and then they proceed by defining their framework for modified Higgs couplings. I am afraid that introducing equation 3 simply as "notation" hides the assumption that the "effective lagrangian" the authors have in mind only includes the tensor structures already present in the SM lagrangian. Incidentally, do the authors plan to include additional interfaces, e.g. to the SMEFT? Or will this be left to the user?

2- In the footnote 9, page 16, the authors mention the Run 1 ATLAS+CMS combination. The current version of the Lilith database (DB 19.06) includes bi-variate gaussian fits to figure 14 of the Run 1 ATLAS+CMS combination paper 1606.02266. There, for each decay mode, an ellipsis is shown in the VBF+VH vs ggF+ttH signal strengths plane. On the other hand, 1606.02266 provides results for a much more generic parametrization, where 20 independent products of cross sections (for ggF, WH, ZH, VBF, ttH) and branching fractions (to gamma gamma, ZZ, WW, tautau, bb) are determined simultaneously. In particular, the best fit values are given in table 8 and the correlation matrix in figure 27. As it is noted in appendix A of that paper, a few significant (anti)correlations are present, that are not captured by the "semi-inclusive" plot in figure 14. Support for multidimensional (variable) gaussian likelihoods has been introduced in the current Lilith version, and was in fact exploited for the implementation of the CMS Run 2 combination of 36/fb data. The authors could comment on their choice not to implement such more general fit to Run 1 combination data, which seems to contradict what stated in the second bullet point in the conclusions.

3- I think that, in the introduction of section 6, the authors could remind to the reader that the Higgs couplings fits they describe rely on specific assumptions and do not represent general, model-independent statements on "the Higgs couplings". The authors could possibly also refer the reader, for completeness, to recent global fits in more general BSM frameworks.

4- It would be interesting to see to what extent the variable-width multi-variate gaussian likelihood (built, for the generic parametrization, with asymmetric uncertainties and with the full correlation matrix) is able to reproduce the asymmetries shown in more constrained experimental likelihoods, as the ones in figure 14 of 1606.02266. Would it be straightforward to derive such "reduced" likelihoods? Would there be a range of signal strengths beyond which artefacts show up?

---

## Round 2 · Author Response

We thank the referees for their positive assessment and the pertinent comments, which indeed helped us improve our paper. Our response to the requested changes is as follows:

Referee 1

1- On p.7 the authors discuss of the implementation of a variable Gaussian and raise the potential issue of singularities of the log-likelihood function. It is not clear from this discussion how the authors deal with these singularities in practice, what the region of validity of the variable Gaussian implementation is, and why using an ordinary two-sided Gaussian instead (outside of the region of validity) is legitimate in the extremely asymmetrical case σ− → 0. The authors might want to comment on this.

Reply: In practice Lilith issues an error when a singularity is hit, e.g. when the user tries to input (or scan over ranges of) reduced couplings beyond the range of validity. Writing the σ− → 0 case as an ordinary two-sided Gaussian (that is two Gaussians, one for the positive side and one for the negative side, put together) in the database is just a trick to avoid computational instabilities, if they occur. It means that only the positive branch of the Gaussian is used to compute the likelihood; for negative signal strength values the likelihood goes to zero in this case. Whether or not this is needed has to be decided case-by-case upon validation. In fact this is not used (not necessary) for the data in the current database; we have included the comment only for completeness since we are discussing the likelihood calculation in general terms, and it may be useful information for users who want to extend the database themselves. In the revised version of the paper, we have slightly modified the sentence about σ− → 0 and added a comment that Lilith issues an error message whenever a numerical problem occurs.

About the region of validity of the variable Gaussian implementation, this is difficult to determine as the exact likelihood function is not available to us. Mathematically, this region is bounded by the condition -2*Log(L) > 0. This however does not guarantee that the approximation is good. One must therefore always validate the variable Gaussian (or any other approximation) implementation against the results provided in the experimental papers, which is what we do.

2- Some of the used analyses have been updated for a larger data set already (see weaknesses). The authors could consider updating the analyses available for a higher luminosity, i.e. tth 1806.00425 (H→γγ, H→ZZ); and VH, H→bb 1808.08238.

Reply: We deliberately used only the 36/fb Run2 results to have a consistent Lilith database for this dataset. From our point of view this is preferable over mixing results from different datasets in global fits. The present work is therefore confined to 36/fb Run2 results, and we wish to stick to this choice. This said, results for larger datasets, notably for the full 150/fb of Run 2, are being implemented as they come out and will be made available on the Lilith-2 master branch on Github once they are validated -- this will eventually lead to a new official release when a large enough set of 150/fb results has been implemented and validated.

3- The authors could consider adding a number of potentially relevant Higgs analyses.

  • H→inv: ATLAS WBF analysis 1809.06682

  • H→μμ: ATLAS 1705.04582

  • VH, H→WW: ATLAS 1903.10052

Reply: We have added these results to what is now DB 19.09, discussed them in Section 5.1 and updated all results in Section 6 accordingly.

Regarding the H→μμ result, we had to make a choice. Concretely, we need to reconstruct the likelihood, -2logL, of the H->mu+mu- signal strength from the information given in the ATLAS study. In the last paragraph on page 6 of the ATLAS paper, it is written that "the measured overall signal strength is μS=−0.1±1.5". We would normally implement this as a Gaussian with mean −0.1 and standard deviation 1.5, i.e. -2logL(mu) = (-0.1-mu)^2/2.25. However, this would give a 95% CL limit of 2.84 (2.36) for a two-sided (one-sided) test, while the 95% CL limit quoted in the ATLAS paper is 3.0. The difference most likely comes from the CLs procedure for the limit setting. Alternatively, one can mimick the 95% CL limit of 3.0 by -2logL(mu) = (0-mu)^2/1.53^2, for a two-sided test. Unfortunately there is no validation material is available for testing the different choices against official ATLAS fits. We have in fact contacted the ATLAS Higgs conveners about the issue but are still waiting for a reply. So, in order to include the ATLAS H→μμ result without too much delaying the publication of our paper, we chose to stick with the more conservative choice of -2logL(mu) = (0-mu)^2/1.53^2. In any case, the impact for the results in Section 6 is very small.

4- Most importantly, the limit on the invisible Higgs branching is interesting, but not obvious. In Fig.12 and the corresponding text on p.18 the authors find a combined 95% CL limit on BR(H→inv) of <= 5% for the SM case. However, the included ATLAS and CMS analyses quote limits of 33% (CMS) and 67% (ATLAS). It is not clear how these limits combine to a 5% limit. The authors should comment on how they obtained this limit and how it compares to the results of the experimental papers.

Reply: In the SM+inv case, BR(H→inv) is primarily constrained from the signal strength measurements in the visible SM final states. We have added a footnote (footnote 12) to explain this.

Referee 2

1- On page 2 the authors describe the assumptions and the scope of the signal strength framework, and then they proceed by defining their framework for modified Higgs couplings. I am afraid that introducing equation 3 simply as "notation" hides the assumption that the "effective lagrangian" the authors have in mind only includes the tensor structures already present in the SM lagrangian. Incidentally, do the authors plan to include additional interfaces, e.g. to the SMEFT? Or will this be left to the user?

Reply: The assumptions and limitations of the signal strength framework are discussed in detail in arXiv:1307.5865, as we mention in footnote 1 on page 2. Moreover, on page 2 we note that "The signal strength framework is based on [...] the assumption that new physics results only in the scaling of SM Higgs processes." This is equivalent to stating that the effective Lagrangian only includes tensor structures already present in the SM Lagrangian. We have added a footnote (footnote 3) where we state this explicitly.

Regarding interfaces to, e.g., SMEFT, this may come in the future if there is user request for it, but it is not a priority at the moment.

2- In the footnote 9, page 16, the authors mention the Run 1 ATLAS+CMS combination. The current version of the Lilith database (DB 19.06) includes bi-variate gaussian fits to figure 14 of the Run 1 ATLAS+CMS combination paper 1606.02266. There, for each decay mode, an ellipsis is shown in the VBF+VH vs ggF+ttH signal strengths plane. On the other hand, 1606.02266 provides results for a much more generic parametrization, where 20 independent products of cross sections (for ggF, WH, ZH, VBF, ttH) and branching fractions (to gamma gamma, ZZ, WW, tautau, bb) are determined simultaneously. In particular, the best fit values are given in table 8 and the correlation matrix in figure 27. As it is noted in appendix A of that paper, a few significant (anti)correlations are present, that are not captured by the "semi-inclusive" plot in figure 14. Support for multidimensional (variable) gaussian likelihoods has been introduced in the current Lilith version, and was in fact exploited for the implementation of the CMS Run 2 combination of 36/fb data. The authors could comment on their choice not to implement such more general fit to Run 1 combination data, which seems to contradict what stated in the second bullet point in the conclusions.

Reply: Indeed we tried to use the best fit values from table 8 and the correlation matrix from figure 27 of Run 1 ATLAS+CMS combination paper 1606.02266. However, with this we cannot reproduce well the official Run 1 coupling fits (from 1606.02266). The original bi-variate gaussian fits to figure 14 perform better in the validation, so we decided to keep them. However, since our paper is about the usage of the Run 2 results, we do not wish to enter into a detailed discussion of this issue; we think it would be more appropriate to discuss this in talks than in this paper.

3- I think that, in the introduction of section 6, the authors could remind to the reader that the Higgs couplings fits they describe rely on specific assumptions and do not represent general, model-independent statements on "the Higgs couplings". The authors could possibly also refer the reader, for completeness, to recent global fits in more general BSM frameworks.

Reply: We expanded the introductory paragraph of section 6 accordingly.

4- It would be interesting to see to what extent the variable-width multi-variate gaussian likelihood (built, for the generic parametrization, with asymmetric uncertainties and with the full correlation matrix) is able to reproduce the asymmetries shown in more constrained experimental likelihoods, as the ones in figure 14 of 1606.02266. Would it be straightforward to derive such "reduced" likelihoods? Would there be a range of signal strengths beyond which artefacts show up?

Reply: Yes, with a bit of work it is possible to derive contours of constant CL in mu(x1,Y) vs. mu(x2,Y) planes, like in figure 14 of 1606.02266. However, given Run 2 precision, x1=ggF+ttH and x2=VBF+VH means a loss of information compared to the full likelihood computed by Lilith, so the contours may not be unique. This, and the question of the range of validity, will need further study, beyond the scope of the present paper.

Editor's request

In addition to implementing the changes requested by the referees, we followed the advise by the editor to "1. specify which SM predictions are actually used in the codes and 2. indicate to which order the new physics effects are included" by adding this information in Section 4. Moreover, for completeness, we now mention at the beginning of Section 6, where we comment on the assumptions made, that "Details about the accuracy of the SM predictions are also provided in Section 4."

---

## Round 2 · List of Changes

- page 2: added footnote 3;
- page 7, paragraph after eq. (5): modified comment on σ− → 0 and added sentence that "Lilith issues an error message whenever a numerical problem occurs.";
- pages 9-10: throughout section4, specified which SM predictions are actually used and to which order they are computed;
- page 10: added introductory paragraph to section 5, explaining the change from DB 19.06 to DB 10.09;
- section 5.1: added discussion of three ATLAS results (HIGG-2017-14, HIGG-2016-10 and HIGG-2018-54); added validation plot for HIGG-2017-14 in Figure 3;
- page 17: expanded introductory paragraph of section 6 to recall assumptions made in the coupling fits (and the overall approach);
- page 17: added references (refs 44-47) to other global fits to Run 2 Higgs results;
- all of section 6: updated all numbers and plots to DB 19.09;
- page 20: added footnote 12;
- references: added refs. 14, 15, 24, 26, 27, and 44-47.

You are currently on this page

Resubmission 1908.03952v2 on 7 October 2019

---

## Editorial Decision

published